# Tackling the Problem of Large Deformations in Deep Learning Based Medical Image Registration Using Displacement Embeddings

**Lasse Hansen**                                             HANSEN@IMI.UNI-LUEBECK.DE

**Mattias P. Heinrich**                                   HEINRICH@IMI.UNI-LUEBECK.DE

*Institute of Medical Informatics, Universität zu Lübeck, Lübeck, Germany*

## Abstract

Though, deep learning based medical image registration is currently starting to show promising advances, often, it still falls behind conventional frameworks in terms of registration accuracy. This is especially true for applications where large deformations exist, such as registration of interpatient abdominal MRI or inhale-to-exhale CT lung registration. Most current works use U-Net-like architectures to predict dense displacement fields from the input images in different supervised and unsupervised settings. We believe that the U-Net architecture itself to some level limits the ability to predict large deformations (even when using multilevel strategies) and therefore propose a novel approach, where the input images are mapped into a displacement space and final registrations are reconstructed from this embedding. Experiments on inhale-to-exhale CT lung registration demonstrate the ability of our architecture to predict large deformations in a single forward path through our network (leading to errors below 2 mm).

**Keywords:** deformable image registration, convolutional neural networks, thoracic CT

## 1. Introduction

Recently, learning based medical image registration has shown great advances in different tasks, but, in contrast to other medical image analysis applications, e.g. segmentation, remains a very challenging problem for deep networks. The majority of recently published methods uses encoder-decoder architectures (like the U-Net) to predict dense displacement fields directly from the input images in an unsupervised setting using similarity metrics (minimizing the objective function similar to conventional iterative registration frameworks) (Balakrishnan et al., 2019) or uses annotated label images to guide the training process (Hu et al., 2018). To deal with large deformations in medical images primarily multilevel strategies and iteratively trained networks were proposed (Eppenhof et al., 2019; de Vos et al., 2019; Hering et al., 2019). Still, while deep networks offer very fast inference times and have the potential to further learn from expert annotations, with errors above 2.2 mm they can not yet compete with accuracies of conventional registration frameworks on challenging thoracic CT benchmarks (below 1 mm (Rühaak et al., 2017)).

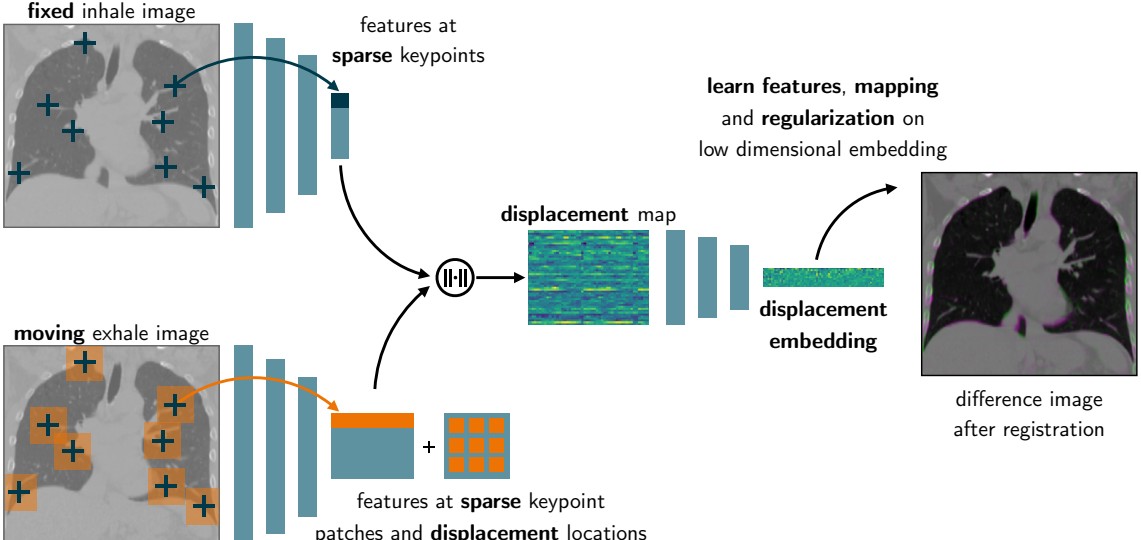

Figure 1: Outline of our proposed architecture for deep learning based medical image registration under large deformations. Dense features are extracted from both, the fixed and moving image, using a convolution neural network. In the fixed image, features are sparsely sampled at given descriptive keypoints and compared with displaced feature patches in the moving image building up a feature displacement map. To make it feasible to learn final predictions and regularized displacements, we propose to map to a (learned) embedding for further processing.

## 2. Methods

In contrast to aforementioned encoder-decoder architectures, in this work we propose to explicitly model the relation between fixed and moving image features with displacement maps. Figure 1 outlines our approach. This concept is similar to the low resolution *correlation layer* for 2d images in FlowNet (Dosovitskiy et al., 2015) and dense volumes in PDDNet (Heinrich, 2019), but we compute the dissimilarities on high resolution feature maps (stride of 2) and therefore employ different strategies to limit the computational burden (this is especially important for learning based approaches as gradients need to be passed through the network in the backward path): 1. we extract fixed features only at sparse keypoints based on the Foerstner interest operator (cf. (Rühaak et al., 2017)), thus reducing the search space and 2. propose to (non-linearly) map the displacement map to a low dimensional embedding space, substantially compressing the displacement features. Further processing, e.g. estimation of final displacements and regularization, is directly employed on the displacement embeddings.

## 3. Experiments and Results

To validate our approach we choose the challenging task of inhale-to-exhale lung registration on the DIR-Lab 4D-CT and DIR-Lab COPD data set (Castillo et al., 2013), as it contains

Table 1: Results for inhale and exhale CT scan pairs of the DIR-Lab 4D-CT and DIR-Lab COPD data set (Castillo et al., 2013), respectively. The mean(standard deviation) target registration error (TRE) in mm is computed on 300 expert annotated landmark pairs per case. *VoxelMorph was trained on affine pre-aligned images using the publicly available code at `https://github.com/voxelmorph/voxelmorph`.

|  | # levels | DIR-Lab 4D-CT | DIR-Lab COPD |
|---|---|---|---|
| initial | – | 8.46(6.58) | 23.36(11.86) |
| (Eppenhof et al., 2019) | 1 | 3.68(3.32) | – |
| DLIR (de Vos et al., 2019) | 3 | 2.64(4.32) | – |
| VoxelMorph*(Balakrishnan et al., 2019) | 2 | 3.65(2.47) | 9.18(4.48) |
| mlVIRNET (Hering et al., 2019) | 3 | 2.19(1.62) | – |
| (Hansen et al., 2019) | 1 | – | 4.30(3.60) |
| ours-256 | 1 | 2.13(1.65) | 4.73(8.56) |
| ours-512 | 1 | **1.97(1.42)** | **3.42(5.63)** |

complex and large deformations. We use a fixed feature extractor (lightweight U-Net with 3 encoder and 2 decoder blocks) that is pretrained to predict MIND-like descriptors (Heinrich et al., 2012) from the input images. Feature patches from the fixed image are sampled at ~1500 distinctive keypoints and compared with feature patches of $21^3 (= 9261)$ voxels at the corresponding locations in the moving image. For this proof-of-concept we simplify our setting and use a PCA embedding (instead of a learned mapping) with 256 and 512 dimensions (thus compressing the displacement space by ~98% and ~95%, respectively). As regularization method, we employ a simple diffusion over all keypoints and displacements using the graph Laplacian. Table 1 shows the results of our method in comparison to other learning based registration frameworks, four approaches based on dense encoder-decoder (multi-level) architectures (Eppenhof et al., 2019; de Vos et al., 2019; Balakrishnan et al., 2019; Hering et al., 2019) and one that is using keypoints with graph CNNs and a point cloud matching algorithm (Hansen et al., 2019).

## 4. Conclusion

We presented a registration framework for large deformations in medical images that, in contrast to recent approaches, explicitly considers a large number of discrete feature displacements and maps them into an embedding space. It outperforms other deep learning based state-of-the-art methods on the DIR-Lab 4D-CT (errors below 2 mm) as well as on the DIR-Lab COPD dataset (errors below 3.5 mm). As this work may be considered as a proof of concept, we see great potential for improvement of our method using a learned non-linearly mapping to the embedding space as well as extending the regularization to use graph CNNs that can learn from the inherent structure of the keypoint graph.

## Acknowledgments

We gratefully acknowledge the support of the NVIDIA Corporation with their GPU donations for this research.

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
