# OpenReview forum: "Tackling the Problem of Large Deformations in Deep Learning Based Medical Image Registration Using Displacement Embeddings"
_MIDL.io/2020/Conference — MIDL 2020_

### Official Review · AnonReviewer4 · 2020-02-26
**Well structured, not uninteresting but currently below MIDL threshold**

**Rating:** 2
**Confidence:** 5

**Review:**

Summary

The authors work on on inhale-to-exhale CT lung registration. They propose a novel approach where a the final registration reconstructed from a (low-dimensional) embedding of the displacement space. Experiments are performed on the DIR-Lab 4D-CT and DIR-Lab COPD data set, including a comparison with five other deep learning based approaches. On the DIR-Lab 4D-CT the proposed method reaches average landmark error of 1.97+-1.42mm.

Strengths

The paper is well structured and easy to read. Evaluation is performed on two established data sets. Figure 1 gives a good impression of the used network without  getting lost in details. The core idea of their work, to explicitly model differences of fixed and moving image features by means of displacement maps and to predict the final registration from an (low-dimensional) embedding of these, is novel.

Weaknesses

>Method:

The proposed method misses the two possibilities to introduce really new and refreshing components by explicitly using the the geometric structure of the keypoint graph for e.g. some graph learning and something more interesting than PCA to achieve the low dimensional embedding. These two points are left for future work.

>Validation:

The validation seems to me a little selective, since I found  three recent papers that employ deep learning methods to perform pulmonary registration on the DIR-Lab 4D-CT data set and that claim a higher accuracy as the proposed one:

1.86+-2.12mm
Sokooti, Hessam, et al. "3D Convolutional Neural Networks Image Registration Based on Efficient Supervised Learning from Artificial Deformations." arXiv preprint arXiv:1908.10235 (2019).

1.66+-1.44mm
Jiang, Zhuoran, et al. "A multi-scale framework with unsupervised joint training of convolutional neural networks for pulmonary deformable image registration." Physics in Medicine & Biology 65.1 (2020): 015011.

1.59+-1.58mm
Fu, Yabo, et al. "LungRegNet: an unsupervised deformable image registration method for 4D‐CT lung." Medical Physics (2020).

If one would consider additionally the non deep learning methods (what is pretty natural from an applicational point of view) one would find way more works that deliver a higher accuracy than the proposed method.

Justification Of Rating

Although the basic idea itself is interesting I don't feel this work is ready to be published now. There is room left for methodological developments that, if carried out, might make this work an interesting contribution to the body of knowledge in the field of pulmonary registration.

The comparison does not cover all the recent deep learning based papers.

Since the the non deep learning methods still deliver superior results (Not to talk about the fact that Rühaak et al. 2017 already reached the accuracy of another  human observer for the DIR-Lab COPD data.), I feel the aspect of computational performance and simplicity should be carried out in more details (and with numbers).

All together I feel this work is not ready yet and I thus opt for weak reject.

---

### Official Review · AnonReviewer2 · 2020-03-14

**Rating:** 3
**Confidence:** 5

**Review:**

Different from existing standard encoder-decoder networks, it proposes an explicit model for correlating fixed (inhale) and moving (exhale) image features. The features are extracted at sparse key points and transformed into a compact representation by CNN. Then a displacement map is calculated to measure their dissimilarity and further represented as displacement embedding.

I think the performance of the proposed method could be affected by the number of key points and the size of the displace locations, although it is understandable not to include all the details due to the page limit. Also, there might be a possibility of estimating unnaturally aggressive deformations since they are estimated based on key points only. In the experiment shown in Table 1, it shows improved performance compared to other deep learning-based methods, including VoxelMorph and less obviously improved result compared to other algorithms for large deformations.

---

### Official Review · AnonReviewer1 · 2020-03-14
**Building on an intriguing previous work**

**Rating:** 2
**Confidence:** 4

**Review:**

This work primarily builds on the idea introduced in Heinrich (2019), namely PDDNet, which itself borrows the correlation layer from FlowNet. The main difference is the use of a non-learned keypoint extractor in order to keep the computational burden in check, the feature dimensionality reduction, and a different graph-based smoothness regularization approach due to the non-regular sampling of keypoints. For feature dimensionality reduction, a simple PCA was employed for this proof-of-concept. Given the closely related work of PDDNet, it is surprising that authors have not compared their adaptation to this method! I would consider it mandatory to demonstrate the benefit of regular vs. non-regular control point distribution. No information at all is provided on how the displacement embeddings are used to predict the final displacement fields. It is very promising work, and I am looking forward to a future conference / journal article on the approach.

---

### Official Review · AnonReviewer3 · 2020-03-19

**Rating:** 3
**Confidence:** 3

**Review:**

This work tackles a challenging problem: inhale-to-exhale CT lung registration.
Based on the work of Heinrich and FlowNet, the authors propose to use a correlation layer to generate a displacement field.

Pros:
- the paper is clear and easy to read
- reducing the computation cost by using keypoints is an interesting approach
- evaluation shows partial but promising results

Cons:
- the authors resort to the Foerstner interest operator. How many keypoints are required?
- there is no comparison with Heinrich's method or other recent deep learning approaches

---

### Meta-Review · Area_Chair1 · 2020-04-07
**MetaReview of Paper155 by AreaChair1**

**Rating:** 3

**Metareview:**

The paper presents an extension of recent work based on keypoint-based CNN registration. All reviewers agree that the work is a incremental improvement over recent method (Heinrich et al, etc), and that a lot more work should be done before the work is mature.

Nevertheless., as this is a short paper, the ideas presented and the thorough comparison make it borderline acceptable for MIDL short paper track. I strongly recommend that the authors address the reviewer worries by the conference time.

**Paper Type:**

methodological development

---

### Decision · Program_Chairs · 2020-04-11

Accept